# Navigating duplication in pharmacovigilance databases: a scoping review

Ronald Kiguba ●,[1] Gerald Isabirye,[2] Julius Mayengo,[2] Jonathan Owiny,[2] Phil Tregunno,[3] Kendal Harrison,[3] Munir Pirmohamed,[4] Helen Byomire Ndagije[2]

[1]Department of Pharmacology and Therapeutics, College of Health Sciences, Makerere University, Kampala, Uganda
[2]National Pharmacovigilance Centre, National Drug Authority, Kampala, Uganda
[3]Safety and Surveillance Group, Medicines and Healthcare Products Regulatory Agency, London, UK
[4]Centre for Drug Safety Science and Wolfson Centre for Personalised Medicine, Institute of Systems, Molecular and Integrative Biology, University of Liverpool, Liverpool, UK

**Correspondence to**
Dr Ronald Kiguba;
kiguba@gmail.com

## ABSTRACT

**Objectives** Pharmacovigilance databases play a critical role in monitoring drug safety. The duplication of reports in pharmacovigilance databases, however, undermines their data integrity. This scoping review sought to provide a comprehensive understanding of duplication in pharmacovigilance databases worldwide.

**Design** A scoping review.

**Data sources** Reviewers comprehensively searched the literature in PubMed, Web of Science, Wiley Online Library, EBSCOhost, Google Scholar and other relevant websites.

**Eligibility criteria** Peer-reviewed publications and grey literature, without language restriction, describing duplication and/or methods relevant to duplication in pharmacovigilance databases from inception to 1 September 2023.

**Data extraction and synthesis** We used the Joanna Briggs Institute guidelines for scoping reviews and conformed with the Preferred Reporting Items for Systematic Reviews and Meta-Analyses Extension for Scoping Reviews. Two reviewers independently screened titles, abstracts and full texts. One reviewer extracted the data and performed descriptive analysis, which the second reviewer assessed. Disagreements were resolved by discussion and consensus or in consultation with a third reviewer.

**Results** We screened 22 745 unique titles and 156 were eligible for full-text review. Of the 156 titles, 58 (47 peer-reviewed; 11 grey literature) fulfilled the inclusion criteria for the scoping review. Included titles addressed the extent (5 papers), prevention strategies (15 papers), causes (32 papers), detection methods (25 papers), management strategies (24 papers) and implications (14 papers) of duplication in pharmacovigilance databases. The papers overlapped, discussing more than one field. Advances in artificial intelligence, particularly natural language processing, hold promise in enhancing the efficiency and precision of deduplication of large and complex pharmacovigilance databases.

**Conclusion** Duplication in pharmacovigilance databases compromises risk assessment and decision-making, potentially threatening patient safety. Therefore, efficient duplicate prevention, detection and management are essential for more reliable pharmacovigilance data. To minimise duplication, consistent use of worldwide unique identifiers as the key case identifiers is recommended alongside recent advances in artificial intelligence.

## STRENGTHS AND LIMITATIONS OF THIS STUDY

⇒ The study involved an extensive literature search on a global scale including both peer-reviewed publications and the grey literature.

⇒ This review adhered to the rigorous methodology published by the Joanna Briggs Institute for scoping reviews.

⇒ We employed an a priori PubMed search strategy, which was then adapted across other peer-reviewed research databases (Web of Science, Wiley Online Library, EBSCOhost) using the Polyglot Search Translator. Additionally, our search for grey literature followed the guidelines outlined in the Canadian Agency for Drugs and Technologies in Health Guide.

⇒ The risk of bias or quality assessment of included studies was not conducted in keeping with the design for scoping reviews, which limited the ability to document the methodological rigour of the included studies.

## INTRODUCTION

Pharmacovigilance is crucial for drug safety as it promotes the prevention, detection and evaluation of suspected adverse reactions to minimise their impact on patient health.[1 2] One of the challenges of pharmacovigilance is the duplication of cases in pharmacovigilance databases, which distorts drug safety and efficacy assessment, and disrupts decision-making.[3–5] Duplication is defined as multiple, unconnected records that refer to the same potential adverse event.[3 4 6] From 2000 to 2010, about 2.5% of reports with adequate information for duplicate analysis in the WHO global pharmacovigilance database were duplicates; the percentage was higher for reports from the literature (11%) and those with fatal outcomes (5%).[4] To promote the accuracy of pharmacovigilance databases, it is important to routinely screen and remove duplicate cases.[4]

The duplication of cases should, however, be differentiated from the replication of records. Replication includes reproducing

or recreating pharmacovigilance reports for various purposes, for example, research, regulatory compliance or data analysis. Replication involves making copies of reports for different stakeholders or systems without necessarily meaning that the content or details of the reports are identical. Replication could result in duplication if multiple reporters, for example, pharmaceutical companies, report the same case to the same regulatory authority for compliance reasons.[7]

Global, regional and country-level guidelines have been implemented to address duplication in pharmacovigilance databases. Global pharmacovigilance initiatives, such as the International Council for Harmonization (ICH) guidelines and the Uppsala Monitoring Centre, which coordinates the WHO Programme for International Drug Monitoring, promote standardisation and harmonisation of pharmacovigilance activities, that are essential for accurate and efficient data collection, analysis and reporting.[2 8–11] In the USA, the Food and Drug Administration (FDA) uses the Adverse Event Reporting System (FAERS) to manage adverse event reports, including duplicates.[12 13] In Europe, the European Medicines Agency oversees the safety of medicines via the EudraVigilance system which uses unique identification numbers for each report and has a built-in deduplication mechanism to identify and eliminate duplicate reports.[14 15] In Asia, a deduplication technique was adopted by the Pharmacovigilance Programme of India, together with the assignment of unique identification numbers and data standardisation. Similar to this, in China, the National Medical Products Administration has put in place a system called the Adverse Drug Reaction Monitoring and Re-evaluation System that incorporates a deduplication mechanism for reporting and monitoring suspected adverse reactions.[16] In Africa, WHO built a regional pharmacovigilance centre in Ghana to support the region's pharmacovigilance systems through training, including duplicate management.[17] Pharmaceutical companies are also responsible for managing duplicate reports. Comprehensive and integrated pharmacovigilance systems are required to effectively manage data from multiple sources, for example, electronic health records, spontaneous reporting databases, social media and digital devices, among others; and to eliminate duplicate reports.[18 19] These systems should ensure compliance with the global pharmacovigilance regulations and standards.

Duplicates arise from various causes including data entry errors,[20 21] multiple reporters and reporting channels,[22 23] pressure from mandatory reporting requirements for healthcare professionals and pharmaceutical companies,[4 24] challenges in data integration and harmonisation in pharmacovigilance databases,[19] multiple representation of the same data due to the merging of pharmacovigilance databases,[6] lack of standardised data management practices,[6] and incomplete case information of key data elements.[12 25]

The implications of duplication are far-reaching, hence the need for routine screening and elimination of duplicates from pharmacovigilance databases.[6] Software programs significantly improve efficiency, accuracy and consistency in the detection and management of duplicates. The Uppsala Monitoring Centre developed the software VigiMatch to detect potential duplicates in VigiBase, the global pharmacovigilance database.[26] VigiBase can merge duplicate reports identified by VigiMatch.[4 27] Automated standalone software programs have been developed to handle large volumes of data and can be customised to meet specific regulatory requirements.[28–30]

Advances in technology, for example, machine learning and natural language processing, hold promise in augmenting the efficiency and precision of pharmacovigilance efforts, particularly in the deduplication of large and complex pharmacovigilance databases.[5 31 32] Deduplication methods are, however, impeded by the high extent of incomplete case information which makes it difficult to conduct reliable duplicate analysis.[4] Moreover, systems ought to avoid over-reliance on automated standalone software programs as they could generate false signals. Instead, holistic methods which include manual review alongside automated processes should be preferred.[33 34] The objective of this scoping review is to provide a comprehensive overview of duplication in pharmacovigilance databases on a global scale. This review addresses the extent, causes, prevention strategies, detection methods, management and impact of duplication in pharmacovigilance databases.

## METHODS

### Research questions

1. What is the extent of duplication in pharmacovigilance databases worldwide?
2. What are the causes of, and prevention strategies, detection methods and management strategies for duplicate reports in pharmacovigilance databases worldwide?
3. What is the impact of duplicate reports in pharmacovigilance databases worldwide?

### Study design

This scoping review was conducted in accordance with the Joanna Briggs Institute (JBI) guidelines for scoping reviews[35] and conformed with the Preferred Reporting Items for Systematic Reviews and Meta-Analyses Extension for Scoping Reviews (PRISMA-ScR).[36] The protocol was developed according to the JBI scoping review template and registered with Open Science Framework (DOI 10.17605/OSF.IO/PBVF3).

### Definition of duplication

Duplication was defined as multiple, unconnected records that refer to the same potential adverse event.[3 4 6]

### Eligibility criteria

The inclusion criteria for selection of articles, reports and guidelines, without language restriction, were as follows:

1. Articles, reports and guidelines related to duplication and/or methods relevant to duplication in pharmacovigilance databases.
2. Articles published in peer-reviewed journals and grey literature from official websites of regulatory authorities and pharmacovigilance organisations.

Articles, reports and guidelines were excluded if, on full-text review, they did not align with the precise objectives of the scoping review. This rigorous screening procedure ensured that only relevant papers were included in the analysis.

### Search strategy

The search strategy was designed to identify all relevant studies related to the research question(s). We therefore searched multiple databases, including PubMed, Web of Science, Wiley Online Library, EBSCOhost, Google Scholar and other relevant websites including regulatory authorities and pharmacovigilance organisations, from inception to 1 September 2023. The reference lists of relevant studies were also screened. The PubMed search string for peer-reviewed publications is detailed in online supplemental appendix 1 and was adapted across the other peer-reviewed research databases using the Polyglot Search Translator.[37] The grey literature search was based on the Canadian Agency for Drugs and Technologies in Health Guide[38] (see online supplemental appendix II).

### Study selection

Two reviewers independently screened the titles and abstracts of all the identified articles for eligibility. Full-text articles were retrieved for all potentially eligible studies and independently assessed for inclusion by two reviewers. Disagreements were resolved by discussion and consensus or in consultation with a third reviewer.

### Data extraction

Data extraction was done by GI using a tool created by the reviewers, as detailed in online supplemental appendix III. This tool was designed to capture various parameters, including the Author, Year of publication, Journal, Country, Title, Specific Objectives, Study design, Categories (extent, prevention, causes, detection, management and impact of duplication in pharmacovigilance databases) and key findings. The extracted data were reviewed by RK and a third reviewer, HBN, and revised accordingly.

### Data analysis and presentation

The extracted data are provided in a descriptive summary according to the review's objectives. The risk of bias or quality assessment was not done across the included studies. This decision aligns with the guidelines outlined by the JBI because a scoping review is not intended to appraise the risk of bias of a cumulative body of evidence.[36] Instead, it focuses on providing a comprehensive overview of the existing literature.

**Table 1** Summary of screening process

| Screening process | Papers (n) |
|---|---|
| Initial screening | 22 745 |
| Full-text review | 156 |
| Eligibility criteria | |
| Met criteria? | |
| Yes | 58 |
| No | 98 |
| Specific categories* | |
| Extent of duplication | 5 |
| Prevention strategies | 15 |
| Causes of duplication | 32 |
| Detection methods | 25 |
| Management strategies | 24 |
| Implications of duplication | 14 |

*The papers overlapped, discussing more than one field.

### Patient and public involvement

Patients and the public did not participate in the design, conduct, reporting and drafting of the dissemination plans of this research.

### Regions and countries

The regions and countries of papers included in this review are presented in online supplemental appendix IV. The papers were from the Asia Pacific region, Europe, Australia and North America.

### RESULTS

We screened 22 745 unique titles and 156 (127 peer-reviewed; 29 grey literature) were eligible for full-text review (see table 1, figure 1 and online supplemental table 1). Of the 156 titles, 58 (47 peer-reviewed; 11 grey literature) fulfilled the inclusion criteria for the scoping review (see table 1, figure 1 and online supplemental table 1). The included titles addressed the extent of duplication (5 papers), prevention strategies (15 papers), causes (32 papers), detection methods (25 papers), management (24 papers) and implications (14 papers) of duplicate reports. The papers overlapped, discussing more than one field. The list of excluded titles is provided in online supplemental appendix V.

### Extent of duplication

The prevalence of duplication in the WHO global pharmacovigilance database, VigiBase, was 2.5% for reports submitted from 2000 to 2010, based on half (51%, 1.9 million of 3.7 million) of the reports with sufficient information for duplicate analysis.[4] Duplication rates were highest for reports from the published literature, for example, peer-reviewed articles (11%), and fatal outcomes (5%).[4] In Europe, the prevalence of duplication in VigiBase was quite high for some countries

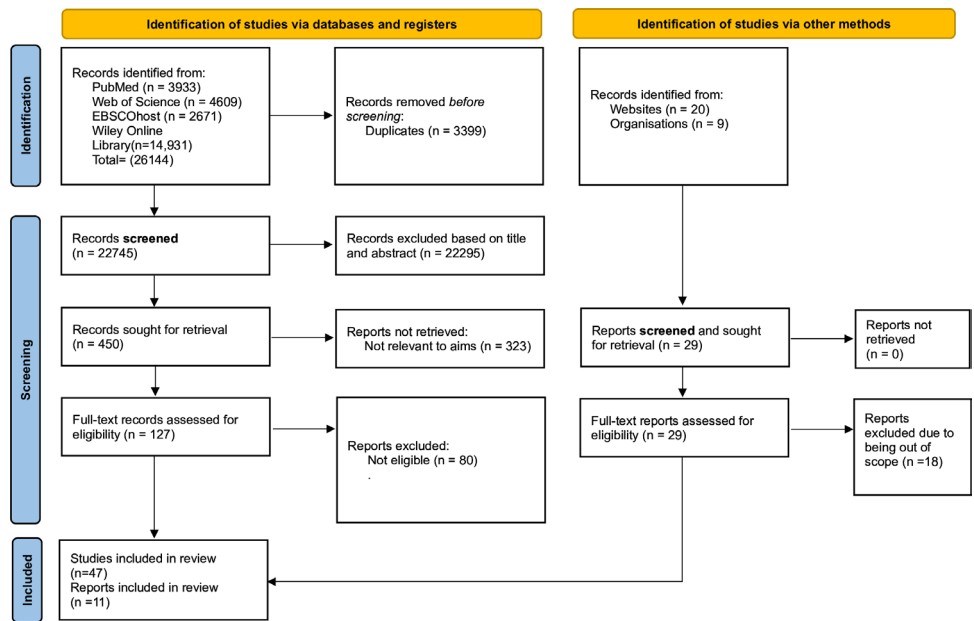

**Figure 1** Preferred Reporting Items for Systematic Reviews and Meta-Analyses flow diagram used to select papers for the scoping review.

(eg, Czech Republic (15%); Austria (15%); Switzerland (4.7%)) and low for others (eg, UK (1.4%); Denmark (1%); Spain (0.7%)).[4] In Asia, the extent of duplication in VigiBase was high in Korea (9.2%) and low in Japan (0.8%). In the USA, duplicates represented 5%–20% of reports in the FAERS by the year 2000.[13] A very high level of duplication, from 66% to 87%, most likely fuelled by replication, was observed in the FAERS database for reports from the literature during the first quarter of 2021.[39] In specific drug-event pair analysis, duplicates accounted for 8% (287/3572) of clozapine-myocarditis cases in VigiBase from inception to January 2021 and for 20% (28/141) of quinine-induced thrombocytopenia cases in the FAERS from 1974 to 2000.[40 41] In Africa, information was scarce on the extent of duplication in pharmacovigilance databases.

### Causes of duplication
Duplicates in pharmacovigilance databases arise from various causes including: (1) data entry errors, for example, (a) submission of follow-up reports of the same case with new identification numbers,[4] (b) multiple entries of the same case by different data entrants[21 42] and (c) misspelling the name of a drug or patient[21]; (2) multiple reporters, for example, patients, healthcare professionals or pharmaceutical companies[10 22 42 43]; (3) pressure from mandatory reporting requirements for healthcare professionals and pharmaceutical companies[4 24 44 45]; (4) challenges in data integration and harmonisation in database software, for example, technical glitches that could occur during (a) data synchronisation, (b) data import/export or (c) the integration of different databases[26 46]; (5) multiple representation of the same data due to the merging of data from different databases such as regulatory authorities, pharmaceutical companies and clinical trials[6]; (6) lack of standardised data management practices characterised by inconsistencies in data entry standards and coding systems[6] and (7) incomplete or missing case information of key data elements, for example, patient identifiers.[25 46 47]

### Strategies to minimise duplication
Several papers described best practices for minimising duplication. These strategies include standardised data entry to enable consistent recording of adverse event data, making it easier to identify duplicates[46 48 49]; assigning unique identifiers to individual case safety reports, such as the worldwide unique case identification number, which should not be changed during electronic transmission and retransmission of safety reports[3 6 50]; regular data cleaning and deduplication[48]; automated and semiautomated search criteria based on similarities in patient demographics (age, gender), unique identifiers, adverse reactions and suspected medicines[4 6 48 51]; data validation and quality assurance processes[52 53]; collaboration between health facilities, pharmaceutical companies and patients[54 55]; and, data sharing between regulatory authorities and international databases.[54 55]

### Detection of duplicates
25 papers discussed the detection of duplicates. Pharmacovigilance databases should be reviewed routinely for duplicates and screening for duplicates should be a priority when new reports are submitted.[6 56] Automated or semiautomated detection systems are recommended to identify duplicates during both manual and electronic data capture.[6 48] To detect duplicates in small databases, reports may be sorted into simple tables by case ID numbers, patient characteristics (age, sex), adverse reaction(s) or suspected medicine(s). For large databases,

however, the flagging of potential duplicates is largely automated and is frequently followed up with manual review particularly to confirm inconclusive duplicates detected by the automated methods.[29] Duplicate detection algorithms should, however, be validated and fine-tuned periodically because the level of detail in submitted reports could vary with time, for example, if data fields become mandatory, these variables ought to be included in the algorithms.[6 26 57]

### Approaches to duplicate detection
#### Exact matching
This is a straightforward method which compares two records to identify precise matches (exact duplicates) based on key data fields such as the report ID, patient demographics, name of adverse event, onset date of the adverse event and name of suspected drug, among others.[58 59]

#### Rule-based matching
This method defines specific conditions to identify and eliminate potential duplicates based on predefined criteria informed by domain knowledge, historical data and observed patterns in the data. It allows for more targeted identification of duplicates based on known patterns, for example, the same case with the same adverse event reported several times in a short period.[60] These rules are typically based on particular fields in the data, for example, patient's name, date of birth or drug name and are intended to match reports that have similar data.[61 62] Simple duplicates can often be found using rule-based systems, but more complex duplicates may be missed because they require more complicated algorithms.[60]

#### Fuzzy matching
This technique identifies records that are similar though not necessarily identical. Unlike exact matching, which requires an exact match between data fields, fuzzy matching allows for variations, misspellings and small differences in the data. This is particularly useful where the data may be inconsistently entered or contain errors, frequently in real-world databases.[63] Fuzzy matching algorithms assess the degree of resemblance between two records using a variety of similarity metrics, including Levenshtein distance and Jaro-Winkler distance. Fuzzy matching algorithms are frequently used in pharmacovigilance to identify duplicates that contain distinct patient names, medical terms or drug names spelt differently or abbreviated differently.[63]

#### Machine learning and natural language processing
This approach employs supervised machine learning models to train algorithms using labelled data that differentiate duplicate and non-duplicate pairs. These models then use the knowledge gained from labelled data to predict whether new record pairs are duplicates or not based on learned patterns.[5 64–66] Natural language processing techniques have been developed to augment computerised duplicate detection using the narrative in adverse event reports.[66] These models can be more precise than rule-based systems, particularly when working with large and complex datasets. The performance of machine learning models can be improved over time as they gain knowledge from fresh instances and adapt to changing data.[18 55] Machine learning models are trained on extensive datasets of adverse event reports to identify patterns and connections between various factors, including drug names, patient demographics and symptoms.[67] Once trained, these models can be used to automatically identify and categorise duplicates based on a variety of factors, including the similarity of medication names, the occurrence of specific medical conditions and the timing and severity of symptoms.[68 69] Machine learning models can be very accurate and efficient in finding complex duplicates; however, their creation and maintenance demand a significant quantity of high-quality training data and specialised knowledge.[70 71]

#### Probabilistic record linkage
This method identifies and removes duplicate reports from large and complex datasets. An example is the hit-miss model which uses computational methods to analyse extensive healthcare databases. It distinguishes 'hits' (adverse events) from 'misses' (non-events) by identifying patterns and signals.[3 48] The method relies on statistical techniques to estimate the probability that two reports refer to the same event or individual, considering various data fields, such as patient name, age and drug name. Probabilistic record linkage is particularly useful when dealing with datasets that are incomplete, inconsistent or have missing data fields because they can still produce reliable results. However, this method can be computationally intensive and requires substantial computational resources, making it less practical for smaller datasets or less well-resourced pharmacovigilance systems.[4]

#### Manual review
This method is used to confirm reports that have been flagged as potential duplicates. Manual review is used in instances where automated methods are inconclusive.[6 48]

### Confirmation of duplicates
Manual confirmation is always necessary following the detection of potential duplicates. Well-documented reports, with elaborate case narratives, make the confirmation process easier.[5 64 66] However, reports with limited information require that reporters are contacted for additional information, which necessitates the timely review of submitted reports.[6] When duplicates are detected, root cause analysis should be done and corrective action taken. Lastly, confirmed duplicates should be managed appropriately.[6]

### Management of duplicates
This is the process of merging two or more reports to form one master report. Two approaches could be used: (1) allocating an existing report as the master report,

with the information from other reports being added unless the same or more precise information is already present in the master report and (2) creating a master report by combining the information from several duplicate reports.[6 21] The master report should always include the case reference numbers of all subordinate duplicate reports for easy traceability. Subordinate duplicates remain in the database for the purpose of audit trail but are not used for any other pharmacovigilance purpose. One challenge with duplicate management is when conflicting information is provided by different reporters. Clarification of conflicting information should be obtained wherever possible.[6]

### Software programs

These are valuable tools for deduplication in pharmacovigilance databases. VigiMatch is a software program developed by the Uppsala Monitoring Centre to detect potential duplicates in the global database of adverse event reports, VigiBase.[26 72] VigiMatch compares various data elements of adverse event reports and assigns a matching score to each pair of reports.[4] Although VigiMatch and VigiBase are separate programs, they are linked in the sense that VigiMatch is often used as a tool to identify and consolidate potential duplicate reports within VigiBase. VigiBase includes data cleaning and deduplication tools that use various criteria to identify potential duplicates, including patient information, drug information and adverse event descriptions. VigiBase can merge duplicate reports identified by VigiMatch, ensuring that the database remains accurate and efficient.[48] VigiMatch is available for use by national pharmacovigilance centres and regulatory authorities worldwide. However, access to the software depends on various factors, such as the availability of funding, infrastructure and technical expertise to effectively implement the software.[72]

Other software packages have been developed. The Medical Dictionary for Regulatory Activities coding system is a standardised software for regulatory reporting that can identify duplicates and minimise data entry errors.[73] ArisGlobal's LifeSphere is a cloud-based pharmacovigilance software with a deduplication module to identify and consolidate duplicate reports.[74] Oracle's Argus Safety is a comprehensive safety management system with a deduplication feature to eliminate duplicate reports and improve data accuracy.[75] Ablebits Ultimate Suite of tools is software with a variety of *add-ins* and *plugins* for Microsoft Office Excel and Google Sheets. The software has a range of tools, including the 'Remove Duplicates' feature that allows users to find and eliminate duplicate entries in a dataset. The primary purpose of Ablebits is to enhance the functionality of spreadsheet programs and make it easier for users to perform various tasks. Ablebits offers a range of tools that can automate repetitive tasks, format data, merge cells, compare data, etc. The software can also merge duplicate reports to create a single case record, thereby reducing manual review and data entry. The goal of Ablebits is to help users save time and improve their productivity while working with Excel or Google Sheets.[76] Open source software for data deduplication, for example, DataCleaner Extension, can be used where Excel has limits.[77]

### Advantages and limitations of software programs

The *advantages* of using software programs for deduplication include improved efficiency, accuracy and consistency in identifying and managing duplicates. These programs can also handle large volumes of data and can be customised to meet specific regulatory requirements.[49] Using Excel with the Ablebits Ultimate Suite of tools could be more cost-effective for small datasets—the software is accessible and easy to use particularly if the user is familiar with Excel. However, Ablebits may not be as efficient or accurate as the dedicated software programs, particularly for large and complex pharmacovigilance databases.

However, there are also *limitations* to using software programs for deduplication. These programs may require extensive training and expertise to operate effectively and might not detect all potential duplicates owing to variations in data entry or differences in data elements used for comparison. Additionally, the cost of implementing and maintaining these software programs may be a barrier for some regulatory authorities, especially in low and middle-income countries (LMICs). Artificial intelligence algorithms need to be trained with large quantities of high-quality data. The technical challenges of artificial intelligence-based pharmacovigilance, particularly in LMICs, are the lack of high-quality databases, insufficient human resources, weak artificial intelligence technology, data sharing and privacy challenges, transparency of algorithms, interoperability across multiple platforms and insufficient support from governments.[28 33 71 78–80]

### Impact of duplication
#### Inaccurate risk assessment

Duplicates in pharmacovigilance databases can lead to overestimation or underestimation of the frequency of reported adverse events, introduce inconsistencies in data analysis and reporting, distort the understanding of the safety profile of medicines and hinder signal detection and risk assessment. This compromises the integrity and reliability of databases as sources of accurate information and potentially impacts important decisions related to drug labelling, risk management and patient safety.[3 4 6 7 13 21 48 64] As an example, Brinker and Beitz[41] observed an exaggerated safety signal of quinine-induced thrombocytopenia as a result of duplicate reports.

#### Resource wastage

Duplicates consume valuable database storage space, computing resources and human effort. Maintaining and managing duplicate records uses up resources, which could be better used for other critical tasks.[4 77 81] Having duplicate entries necessitates additional manual effort to identify and resolve them. Pharmacovigilance staff need

to spend more time on data cleaning and deduplication processes, which diverts their attention from other essential activities.[3]

### Delays in data analysis

Sorting through duplicate entries to identify and remove them can cause delays in data analysis and signal detection. Timely identification and response to safety concerns may be hindered due to the time-consuming nature of duplicate management.[3]

### Regulatory compliance challenges

Duplicates can complicate regulatory reporting and compliance obligations. Thus, submitting accurate and non-duplicated data to regulatory authorities may become more difficult and time-consuming, particularly for pharmaceutical companies.[6]

## DISCUSSION

This scoping review sought to evaluate the extent, causes, preventive strategies, detection methods, management and impact of duplication in pharmacovigilance databases worldwide. The extent of duplication in pharmacovigilance databases varied across different databases. The WHO global database, VigiBase, exhibited a 2.5% prevalence of duplication for reports submitted from 2000 to 2010, based on only half of the reports with sufficient information to conduct duplicate analysis.[4] In a global context, despite that this prevalence rate is relatively low, it represents about 750 000 duplicates among the 30+ million reports in VigiBase as of 2023.[82] Duplication rates in VigiBase were highest for reports from the published literature, for example, peer-reviewed articles (11%), and fatal outcomes (5%)[4]; and could be due to the heightened awareness and documentation of certain types of adverse events, particularly serious outcomes and those of special interest.[4 39] These prevalence rates in VigiBase should, however, be interpreted with caution since half the reports were excluded from duplicate analysis due to missing data.[4] Duplication was more extreme (66%–87%) for individual case safety reports from the published literature in the United States' FAERS database and is almost entirely attributable to the replication of case reports due to the obligation to report individual case safety reports from the literature as enforced on pharmaceutical companies by Competent Authorities.[39] Thus, replication of case reports is an adjunct to duplication and is largely induced by regulatory reporting obligations, contractual agreements and partnerships leading to the creation of case replicas.[7] The resolution of replication will require significant care and possibly redesigning pharmacovigilance data management systems at the global level.

Duplication rates in VigiBase varied across countries, being high for the Czech Republic (15%), Austria (15%), Korea (9.2%) and Switzerland (4.7%) and much lower for the UK (1.4%), Denmark (1%), Spain (0.7%) and Japan (8%), among others.[4] Thus, special attention to data quality is required when integrating pharmacovigilance datasets from different countries with varying risks of duplication. Also, regional variations call for standardised deduplication strategies to ensure consistency and data reliability across different settings.

Duplication in pharmacovigilance databases can arise from data entry errors, multiple reporters, mandatory reporting requirements, technical software glitches and incomplete case information. This is compounded by the growing volume of pharmacovigilance reports worldwide, particularly due to the large number of adverse event reports following mass immunisation with COVID-19 vaccines.[83] Thus, the possibility of duplicates cannot be ruled out, and it is crucial to develop robust and efficient systems to minimise them.[5] Understanding the causes of duplication is important to identify and institute effective measures to prevent them. Such measures could reduce the occurrence of duplication, promote coordinated approaches to deduplication and enhance data quality and consistency. Standardised data entry and the assignment of unique identifiers to individual case safety reports are essential for consistent recording of adverse event data, making it easier to minimise and manage duplicates. The International Council for Harmonization of Technical Requirements for Pharmaceuticals for Human Use has adopted the ICH E2B(R3) messaging standard for electronic transmission of individual case safety reports.[50] This standard stipulates the use of a worldwide unique case identification number (Section C.1.8.1) for each safety report which should not be changed during the transmission and subsequent retransmission of safety reports.[50] This single field becomes the primary key for case identification and is therefore a practical solution to effectively reduce and manage duplication. Regular automated and semiautomated data deduplication mechanisms, leveraging artificial intelligence, are among the array of measures at various stages of deployment across the USA, Europe and Asia.[4 6 48 51] However, there is notable scanty information from Africa. The effectiveness of deduplication methods will require continuous improvement, with time, to be in tandem with the submission of new reports to the pharmacovigilance databases.[5 48 64]

To enhance the accuracy of pharmacovigilance databases, newly submitted adverse event reports should be screened for duplicates. This proactive approach to data entry ensures superior data quality compared with relying solely on postentry data cleaning processes.[6] Approaches to duplicate detection range from manual review to automated methods such as exact matching, rule-based matching, fuzzy matching, probabilistic record linkage and artificial intelligence. Probabilistic record linkage is effective for dealing with incomplete or inconsistent data but demands computational resources.[4] Artificial intelligence models based on machine learning provide more precise and efficient means of identifying duplicates, especially in large and complex datasets.[18 55] Natural language processing techniques which use the narrative in adverse event reports have also been developed to

augment duplicate detection using machine learning. These advances in natural language processing hold promise in enhancing the efficiency and precision of the deduplication of large and complex pharmacovigilance databases.[66] Automated detection systems are the recommended methods for large and complex databases.[5 31 32] However, manual review is necessary for inconclusive potential duplicates from automated systems.[6 29 48] The duplicate detection algorithms of automated systems require routine fine-tuning to adapt them to changing data requirements, ensuring database accuracy.[18 55]

Software programs for the deduplication of pharmacovigilance databases significantly enhance the quality of pharmacovigilance data and ultimately improve patient safety. Dedicated software programs for duplicate management are suitable for very large databases and might not be cost-effective for small databases, particularly in LMIC.[26 73–75] Additional challenges for implementing dedicated deduplication software in LMIC include poor data quality, limited resources and infrastructure, inadequate training and capacity building, a lack of harmonisation and standardisation across regulatory systems and limited access to data and information. These gaps can hinder the ability of regulatory authorities in LMIC to effectively monitor and assess drug safety and efficacy, and the impact of duplication on pharmacovigilance databases.[79] Alternative deduplication software for small datasets in LMIC includes the Ablebits Ultimate Suite of tools with *add-ins* for Microsoft Office Excel. Using Ablebits in addition to manual review for small datasets could be cost-effective and possibly improves the efficiency and accuracy of deduplication.[76] However, open source software solutions, for example, DataCleaner Extension, have superior capabilities to the Ablebits Excel *add-in*.[77]

Duplicates have consequences for pharmacovigilance databases such as the inaccurate risk assessment which could lead to overestimation or underestimation of adverse event frequencies. This could, in turn, distort the understanding of a drug's safety profile and hinder signal detection.[3–5] Resource wastage, increased workload for pharmacovigilance staff, delays in data analysis and regulatory compliance challenges are additional consequences of duplication.[3 4 6 7 77 81] These consequences underscore the importance of addressing duplication to promote the integrity and effectiveness of pharmacovigilance systems. We therefore make the following recommendations:

► We strongly advocate for the consistent use of worldwide unique identifiers (field C.1.8.1 within the ICH E2B(R3) guideline) as the primary key for case identification to minimise case duplication.
► Scale up reporting methods that minimise incomplete case information, for example, electronic systems that enforce the obligatory inclusion of key case details at the time of reporting.
► Increase investment in pharmacovigilance infrastructure and capacity building.
► Improve collaboration and data sharing among regulatory authorities and industry partners.

► Put more emphasis on the harmonisation and standardisation of regulatory systems worldwide.
► Develop and implement innovative technologies such as machine learning and natural language processing to improve the efficiency and accuracy of drug safety monitoring, particularly in LMIC.
► LMICs with small pharmacovigilance datasets ought to conduct pilot studies to explore the feasibility of alternative low-cost technologies for data deduplication (eg, Ablebits Suite of tools in Microsoft Office Excel and the open source DataCleaner Extension).

This review has important limitations. First, the review did not evaluate the risk of bias or quality of included studies, in keeping with the design for scoping reviews, which limited the ability to document the methodological rigour or validity of included studies. Second, the majority of included papers were from Europe and North America which could have affected the generalisability of findings to other geographical regions. These limitations notwithstanding, however, the review provides a comprehensive overview of the cumulative body of literature on duplication in pharmacovigilance databases worldwide.

## CONCLUSION

This scoping review provides a comprehensive overview of duplication in pharmacovigilance databases worldwide. It highlights the need for standardised practices, advanced detection methods and collaborative efforts to minimise duplication and enhance the accuracy and reliability of drug safety data. Addressing duplication is crucial to supporting evidence-based decision-making in healthcare and drug regulation, ensuring patient safety. Software programs are commonly used for deduplication in pharmacovigilance databases and they significantly enhance the quality of pharmacovigilance data and ultimately improve patient safety. Overall, software programs offer advantages in terms of efficiency and accuracy, although there are limitations. For smaller datasets, Microsoft Office Excel with Ablebits Ultimate Suite or open source DataCleaner Extension, among others, may be cost-effective and ought to be piloted, whereas larger and more complex datasets require dedicated software programs.

**Acknowledgements** We are grateful to the National Drug Authority—Uganda, particularly the National Pharmacovigilance Centre, which supported the situation analysis of duplication in pharmacovigilance databases worldwide.

**Contributors** RK and GI conceptualised the study, collected the data, performed data curation, conducted formal analysis and data interpretation and developed the first draft of the manuscript. JM, JO, PT, KH, MP and HBN contributed to the study design, analysis and interpretation of the data. RK, as guarantor, accepts full responsibility for the work and/or the conduct of the study, had access to the data and controlled the decision to publish. All authors read and approved the final draft of the manuscript.

**Funding** This study was funded by the Medical Research Council (MR/V03510X/1).

**Competing interests** RK and MP obtained funding from the Medical Research Council; however, the funder had no role in drafting this manuscript. RK, GI, JM, JO, PT, KH and HBN declare that they have no conflicts of interest. MP has received partnership funding for the following: Medical Research Council Clinical

Pharmacology Training Scheme (co-funded by Medical Research Council and Roche, UCB, Eli Lilly and Novartis). He also has unrestricted educational grant support for the UK Pharmacogenetics and Stratified Medicine Network from Bristol-Myers Squibb. He has developed an HLA genotyping panel with MC Diagnostics but does not benefit financially from this. He is part of the IMI Consortium ARDAT (www.ardat.org). None of this funding MP received is related to the current paper.

**Patient and public involvement** Patients and/or the public were not involved in the design, or conduct, or reporting, or dissemination plans of this research.

**Patient consent for publication** Not applicable.

**Ethics approval** Not applicable.

**Provenance and peer review** Not commissioned; externally peer reviewed.

**Data availability statement** All data relevant to the study are included in the article or uploaded as supplementary information.

**ORCID iD**
Ronald Kiguba http://orcid.org/0000-0002-2636-4115

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
