## [Reviewer comments · BMJ Open]

ARTICLE DETAILS

TITLE (PROVISIONAL)	Navigating Duplication in Pharmacovigilance Databases: A Scoping Review
AUTHORS	Kiguba, Ronald; Isabirye, Gerald; Mayengo, Julius; Owiny, Jonathan; Tregunno, Phil; Harrison, Kendal; Pirmohamed, Munir; Ndagije, Helen

VERSION 1 – REVIEW

REVIEWER	Lewis, David Novartis Pharma GmbH, Patient Safety & Pharmacovigilance I am a full-time employee of Novartis Pharma GmbH. I own stock in Alcon, GlaxoSmithKline, Novartis, and Sandoz, and stock options in Novartis. I am a joint recipient of a grant from the Innovative Medicines Initiative (co-funded by EFPIA and the European Commission). The grant applies to the ConcePTION consortium and provides support only to the public partners. I am Visiting Professor, Department of Pharmacy, Pharmacology & Postgraduate Medicine, University of Hertfordshire, England. This is an honorary position. I am also affiliated to EU2P, a training programme in pharmacovigilance and pharmacoepidemiology with the University of Hertfordshire contributing to taught courses.
REVIEW RETURNED	08-Dec-2023

GENERAL COMMENTS	This manuscript provides a comprehensive review of the occurrence of duplication of individual case safety reports in pharmacovigilance. It is well-written, balanced and offers a valuable update on this important topic. I offer the following comments, which I hope are constructive and are suggestions for improvements: 1. General comment: The manuscript requires careful proof-reading, as there are typographical errors (e.g. on page 6, line Stud should read Study, and there are other minor errors including inappropriate double-spaced text). May I also request that the authors use British (UK) English spelling conventions, as some US English spellings are present. I have attached a PDF file with the majority of the errors marked-up.2. Introduction: Please provide a definition of duplication of ICSRs and add context by highlighting the differences between duplication and replication of ICSRs. Reference 78 provides important insights on the distinction between these two areas.3. Extent of duplication (page 8, lines 11 to 22): The authors should clarify the distinction between duplication and replication in this section. For example, it seems clear that the 'very high level of
--

	duplication, from 66% to 87%, was observed in the FAERS database for reports from the literature' is almost entirely due to replication enforced by the obligations imposed by the Competent Authorities in MAHs reporting ICSRs from the literature. Equally notably (at lines 12 &13), the reference to duplication 'being highest for reports from published literature (11%)' in VigiBase is a direct reflection of the same phenomenon. 4. Discussion: The section requires updating based on the recognition of replication of case reports as an adjunct to duplication. As indicated above, replication is largely induced by regulatory reporting obligations, contractual agreements, and partnerships leading to the creation of case replicas. Resolution of this problem will require significant care, and the potential redesign of pharmacovigilance data management systems at a global level. The authors should emphasize the importance of adhering to the principles set down in ICH E2B(R3) Section C.1.8 concerning worldwide unique case identification. Provision has been made for the population of the field C.1.8.1 'Worldwide unique case identification number'. This is an important addition to the manuscript, as this single field (C.1.8.1) becomes, in effect, the primary key for case identification, and therefore a practical solution to reduce and manage both duplication and replication. Further discussion of potential solutions is required. For example, please consider expanding on one or more of the following themes:  - Standardized data entry to enable consistent recording of adverse event data, making it easier to identify duplicates [ICH E2B(R3), ICH E2D(R1)] - Regular data cleaning and deduplication - Assigning unique identifiers to patients and adverse events - Automated and semi-automated search criteria based on similarities in patient demographics (age, gender), unique identifiers, adverse reactions, and suspected medicines - Improved data validation and quality assurance processes - Collaboration between health facilities, pharmaceutical companies, and reporters - Improved data sharing between regulatory authorities and international databases 5. Recommendations: The authors should consider strong recommendations concerning the use of the field C.1.8.1 (as defined within the ICH E2B(R3) guideline). If this is used as the primary key for case identification, it becomes the pragmatic solution to both reduce and manage case duplication and replication. 6. Table 2 contains many typographical errors and is poorly formatted. It also contains two overtly promotional claims (p38, lines 6 to 9) concerning a commercial database product, and a pharmaceutical company. This text should be removed if it is intended to published Table 2 as an annex.
--	--

REVIEWER	Simmering, Jacob
----------	------------------

	The University of Iowa College of Pharmacy
REVIEW RETURNED	18-Dec-2023

GENERAL COMMENTS	Kiguba and coauthors present a paper summarizing the frequency of duplication - repeated entry of the same adverse drug event - in multiple pharmacovigilance data resources as well as a very high-level overview of how data duplication can be addressed. Recommendations based on this review are efforts to improve data entry accuracy (better data entry will always outperform data cleaning), checking at the time of data entry for data duplication, and a few tools (like Ablebits) that may be suitable for automated deduplication. I am sympathetic to the problem of data duplication. As a health services researcher using administrative health records finding and resolving duplication is a significant part of my professional work. I understand that data duplication can make estimating incidence - and therefore risk - difficult but the other limitations (cost, resource wastage) seem minor compared to the costs and complexity of a program like those run by the FDA, EMA, or WHO. A little more detail on *why* data duplication is problematic in the introduction and conclusion would perhaps “sell” me more on the critical importance of this idea. Specific comments: - Major  1. Need more detail on how the papers defined duplication in your results. 2. I’m not sure what “reports in the published literature” means? Duplication rates are higher but I have no idea what it means (see page 11, lines 10-14) 3. It appears duplication rates also vary by medication, at least in the FDA data. Is this the case and why might it be the case? - Moderate  1. The introduction starts talking about the incidence of duplication and how duplication is a problem before defining duplication. Please define duplication (which you do on page 4, lines 9-10) earlier. 2. I am not sure what is the point of the second paragraph of the introduction (describing pharmacovigilance programs run by the FDA, WHO, EMA, and other agencies). This doesn’t seem to tie into the overall paper. - Minor/typographical/grammar  1. Methods: “Research Question” (singular) should probably be “Research Questions” (plural) to match the three questions described below. 2. Methods “Stud Design” should probably be “Study Design”
---

VERSION 1 – AUTHOR RESPONSE

Reviewer 1

Comment 1: General comment: This manuscript provides a comprehensive review of the occurrence of duplication of individual case safety reports in pharmacovigilance. It is well-written, balanced and offers a valuable update on this important topic.

Response 1: Thank you.

Comment 2: General comment: The manuscript requires careful proof-reading, as there are typographical errors (e.g. on page 6, line Stud should read Study, and there are other minor errors including inappropriate double-spaced text). May I also request that the authors use British (UK) English spelling conventions, as some US English spellings are present.

Response 2: Done. Thank you.

Comment 3: Introduction: Please provide a definition of duplication of ICSRs and add context by highlighting the differences between duplication and replication of ICSRs. Reference 78 provides important insights on the distinction between these two areas.

Response 3: Done. See page 5 of the proposal with track changes. Thank you.

Comment 4: Extent of duplication (page 8, lines 11 to 22): The authors should clarify the distinction between duplication and replication in this section. For example, it seems clear that the 'very high level of duplication, from 66% to 87%, was observed in the FAERS database for reports from the literature' is almost entirely due to replication enforced by the obligations imposed by the Competent Authorities in MAHs reporting ICSRs from the literature. Equally notably (at lines 12 &13), the reference to duplication 'being highest for reports from published literature (11%)' in VigiBase is a direct reflection of the same phenomenon.

Response 4: The manuscript has been updated. See page 10. Thank you.

Comment 5: Discussion: The section requires updating based on the recognition of replication of case reports as an adjunct to duplication. As indicated above, replication is largely induced by regulatory reporting obligations, contractual agreements, and partnerships leading to the creation of case replicas. Resolution of this problem will require significant care, and the potential redesign of pharmacovigilance data management systems at a global level.

Response 5: The manuscript has been updated. See page 14. Thank you.

Comment 6: Discussion: The authors should emphasize the importance of adhering to the principles set down in ICH E2B(R3) Section C.1.8 concerning worldwide unique case identification. Provision has been made for the population of the field C.1.8.1 'Worldwide unique case identification number'. This is an important addition to the manuscript, as this single field (C.1.8.1) becomes, in effect, the primary key for case identification, and therefore a practical solution to reduce and manage both duplication and replication.

Response 6: The manuscript has been updated, see pages 14-15. Thank you.

Comment 7: Discussion: Further discussion of potential solutions is required. For example, please consider expanding on one or more of the following themes: Standardized data entry to enable consistent recording of adverse event data, making it easier to identify duplicates [ICH E2B(R3), ICH E2D(R1)]; Regular data cleaning and deduplication; Assigning unique identifiers to patients and adverse events; Automated and semi-automated search criteria based on similarities in patient demographics (age, gender), unique identifiers, adverse reactions, and suspected medicines; Improved data validation and quality assurance processes; Collaboration between health facilities, pharmaceutical companies, and reporters; Improved data sharing between regulatory authorities and international databases

Response 7: The manuscript has been updated. See page 14-15. Thank you.

Comment 8: Recommendations: The authors should consider strong recommendations concerning the use of the field C.1.8.1 (as defined within the ICH E2B(R3) guideline). If this is used as the primary key for case identification, it becomes the pragmatic solution to both reduce and manage case duplication and replication.

Response 8: Done. See page 15. Thank you.

Comment 9: Table 2 contains many typographical errors and is poorly formatted. It also contains two overtly promotional claims (p38, lines 6 to 9) concerning a commercial database product, and a pharmaceutical company. This text should be removed if it is intended to be published as an annex.

Response 9: Done. Thank you.

Reviewer 2

Comment 10: Kiguba and coauthors present a paper summarizing the frequency of duplication - repeated entry of the same adverse drug event - in multiple pharmacovigilance data resources as well as a very high-level overview of how data duplication can be addressed. Recommendations based on this review are efforts to improve data entry accuracy (better data entry will always outperform data cleaning), checking at the time of data entry for data duplication, and a few tools (like Ablebits) that may be suitable for automated deduplication.

Response 10: Noted with thanks.

Comment 11: I am sympathetic to the problem of data duplication. As a health services researcher using administrative health records finding and resolving duplication is a significant part of my professional work. I understand that data duplication can make estimating incidence - and therefore risk - difficult but the other limitations (cost, resource wastage) seem minor compared to the costs and complexity of a program like those run by the FDA, EMA, or WHO. A little more detail on *why* data duplication is problematic in the introduction and conclusion would perhaps "sell" me more on the critical importance of this idea.

Response 11: Done. See pages 5 & 16. Thank you.

Comment 12: Need more detail on how the paper defined duplication in your results.

Response 12: Done. See pages 5 & 6. Thank you.

Comment 13: I'm not sure what "reports in the published literature" means? Duplication rates are higher but I have no idea what it means (see page 11, lines 10-14)

Response 13: This has been clarified. Thank you.

Comment 14: It appears duplication rates also vary by medication, at least in the FDA data. Is this the case and why might it be the case?

Response 14: We might not be able to ascertain comparability given that the duplication rates are from different reporting periods, countries and databases, being from inception to January 2021 in VigiBase for clozapine-myocarditis cases and from 1974 to 2000 in the United States' FAERS for quinine-induced thrombocytopenia cases.

Comment 15: The introduction starts talking about the incidence of duplication and how duplication is a problem before defining duplication. Please define duplication (which you do on page 4, lines 9-10) earlier.

Response 15: Done. Thank you.

Comment 16: I am not sure what is the point of the second paragraph of the introduction (describing pharmacovigilance programs run by the FDA, WHO, EMA, and other agencies). This doesn't seem to tie into the overall paper.

Response 16: An introductory sentence to the paragraph is added. See page 5. Thank you.

Comment 17: Methods: “Research Question” (singular) should probably be “Research Questions” (plural) to match the three questions described below.

Response 17: Revised. Thank you.

Comment 18: Methods “Stud Design” should probably be “Study Design”

Response 18: Revised. Thank you.

VERSION 2 – REVIEW

REVIEWER	Lewis, David Novartis Pharma GmbH, Patient Safety & Pharmacovigilance I am a full-time employee of Novartis and own stocks in Novartis, Alcon, Sandoz, and GlaxoSmithKline. I am a partner in the IMI ConcePTION project but I do not receive any grant funding. I am Visiting Professor, School of Life and Medical Sciences, at the University of Hertfordshire. This is an honorary position.
REVIEW RETURNED	28-Mar-2024

GENERAL COMMENTS	Thank you for confirming the revisions, I congratulate you on the revised manuscript describing your research. I am very happy to recommend acceptance of a carefully planned, and well executed research study.
--

REVIEWER	Simmering, Jacob The University of Iowa College of Pharmacy
REVIEW RETURNED	01-Apr-2024

GENERAL COMMENTS	My comments have been addressed. Thank you.
---

VERSION 2 – AUTHOR RESPONSE